# A microRNA Signature for the Diagnosis of Statins Intolerance

**DOI:** 10.3390/ijms23158146

**Published:** 2022-07-24

**Authors:** Alipio Mangas, Alexandra Pérez-Serra, Fernando Bonet, Ovidio Muñiz, Francisco Fuentes, Aurora Gonzalez-Estrada, Oscar Campuzano, Juan Sebastian Rodriguez Roca, Elena Alonso-Villa, Rocio Toro

**Affiliations:** 1Research Unit, Biomedical Research and Innovation Institute of Cadiz (INiBICA), Puerta del Mar University Hospital, 11009 Cádiz, Spain; alipio.mangas@uca.es (A.M.); elena.alonso@gm.uca.es (E.A.-V.); 2Medicine Department, School of Medicine, University of Cadiz, 11002 Cádiz, Spain; 3Lipid and Atherosclerotic Unit, Internal Medicine Department, Puerta del Mar University Hospital, 11009 Cadiz, Spain; jsrguezroca@gmail.com; 4Cardiology Service, Hospital Josep Trueta, University of Girona, 17007 Girona, Spain; aperez@idibgi.org (A.P.-S.); ocampuzano@idibgi.org (O.C.); 5Cardiovascular Genetics Center, University of Girona-IDIBGI, 17007 Girona, Spain; 6UCERV, UCAMI, Servicio de Medicina Interna, Hospital Virgen del Rocío, 41013 Seville, Spain; ovidiomuniz@gmail.com (O.M.); aglezestrada@gmail.com (A.G.-E.); 7Lipid and Atherosclerosis Unit, IMIBIC/Hospital Universitario Reina Sofía/Universidad de Córdoba, 14004 Córdoba, Spain; ffuentes@uco.es; 8Centro de Investigación Biomédica en Red, Fisiopatologia Obesidad y Nutricion (CIBEROBN), Instituto de Salud Carlos III, 28029 Madrid, Spain; 9Centro de Investigación Biomédica en Red, Enfermedades Cardiovasculares (CIBERCV), 28029 Madrid, Spain

**Keywords:** circulating microRNAs, statin intolerance, biomarkers, atherosclerotic cardiovascular diseases, statins-adverse myalgia symptoms

## Abstract

Atherosclerotic cardiovascular diseases (ASCVD) are the leading cause of morbidity and mortality in Western societies. Statins are the first-choice therapy for dislipidemias and are considered the cornerstone of ASCVD. Statin-associated muscle symptoms are the main reason for dropout of this treatment. There is an urgent need to identify new biomarkers with discriminative precision for diagnosing intolerance to statins (SI) in patients. MicroRNAs (miRNAs) have emerged as evolutionarily conserved molecules that serve as reliable biomarkers and regulators of multiple cellular events in cardiovascular diseases. In the current study, we evaluated plasma miRNAs as potential biomarkers to discriminate between the SI vs. non-statin intolerant (NSI) population. It is a multicenter, prospective, case-control study. A total of 179 differentially expressed circulating miRNAs were screened in two cardiovascular risk patient cohorts (high and very high risk): (i) NSI (*n* = 10); (ii) SI (*n* = 10). Ten miRNAs were identified as being overexpressed in plasma and validated in the plasma of NSI (*n* = 45) and SI (*n* = 39). Let-7c-5p, let-7d-5p, let-7f-5p, miR-376a-3p and miR-376c-3p were overexpressed in the plasma of SI patients. The receiver operating characteristic curve analysis supported the discriminative potential of the diagnosis. We propose a three-miRNA predictive fingerprint (let-7f, miR-376a-3p and miR-376c-3p) and several clinical variables (non-HDLc and years of dyslipidemia) for SI discrimination; this model achieves sensitivity, specificity and area under the receiver operating characteristic curve (AUC) of 83.67%, 88.57 and 89.10, respectively. In clinical practice, this set of miRNAs combined with clinical variables may discriminate between SI vs. NSI subjects. This multiparametric model may arise as a potential diagnostic biomarker with clinical value.

## 1. Introduction

Cardiovascular disease (CVD) remains the leading cause of mortality and morbidity in Europe [1]. Plasmatic low-density lipoprotein (LDLc) levels is the main causal factor of these vascular events. Statins are the cornerstone of atherosclerotic cardiovascular disease (ASCVD) prevention worldwide. Based on the evidence of the clinical trials, the European Society of Cardiology (ESC)/European Atherosclerosis Society (EAS) guidelines have established a goal of 50% reduction in LDLc concentration in high and very high cardiovascular risk patients to reduce vascular events [1,2]. Most of the trials have demonstrated a remarkable reduction of 22% of major vascular events after five years of statin treatment [3]. Even so, almost 80% of treated patients do not achieve recommended LDLc levels. This low adherence to statin therapy is due to several causes, including cultural reasons, costs, side effects and lack of effectiveness information [4].

Although statins are well tolerated, the major cause of non-adherence to treatment is its adverse effects. The European Medicine Agency defines statin intolerance as the inability to tolerate at least two or more statins at the lowest approved daily dose due to the development of side effects, that began or increase during statin therapy and stop when statin treatment was discontinued [5]. Among these side effects, statin-associated muscle symptoms (SAMS) are the most prevalent reason for the non-adherence and/or discontinuation of statin treatment [6]. Four SAMS forms have been described, namely myalgia, myopathy, myositis and rhabdomyolysis. Rhabdomyolysis is the most severe form of statin-induced muscle damage. Nevertheless, myalgia—muscle complaints without creatine kinase elevation or major functional loss—is usually self-reported in the range of 7–29% [7]. The mechanisms involved in the pathogenesis of SAMS are still unknown. The lack of a gold standard test or biomarker to diagnose this adverse effect, the inconsistent definition of this reality in clinical practice, and the impact on the cardiovascular event rates after dropping this therapy demonstrate the importance of identifying a novel and accessible biomarker that distinguishes this population [8].

MicroRNAs (miRNAs) are small non-coding RNAs identified as important regulators of genes involved in several biological processes [9,10,11]. miRNAs regulate gene expression by degrading messenger RNAs (mRNAs), repressing protein synthesis, or interacting with long non-coding RNAs. Their properties make them the most widely studied extracellular RNAs as diagnostic and therapeutic-tailored markers in the cardiovascular field [12,13,14]. Several studies have described the relationship between statins and miRNA profiles [15,16,17,18], but this is the first time miRNAs have been considered as an alternative diagnostic tool related to statin intolerance detection. Therefore, we aim to determine the role of miRNA as a peripheral biomarker to identify the statin intolerant (SI) population and thus personalize dyslipidemia treatment in these patients at cardiovascular risk.

## 2. Results

### 2.1. Clinical Parameters between SI and NSI Patients

The anthropometrical and clinical features of the SI and NSI cohorts are shown in Table 1.

This study included 84 patients, 39 SI patients and 45 NSI patients as the control group. Considering the groups, the male population were larger than the female population in NSI, but SI female (61.5%) were significantly more frequent (*p* = 0.01); in contrast, a lower percentage of females (31%) was observed in the NSI group. There were no significant differences in terms of age. The presence of ASCVD, the years of dyslipidemia (DLP) (*p* = 0.003, respectively), and high blood pressure (*p* = 0.006) were significantly different between cohorts. It is important to note that non-HDLc plasmatic concentration (*p* < 0.001) displayed significant differences between SI and NSI groups. When considering the medication intake, the use of angiotensin agonist receptors (*p* = 0.004), diuretic (*p* = 0.003), beta-blockers (*p* < 0.001), alpha-blockers (*p* = 0.03) and aspirin (*p* < 0.001) was significantly different between SI and NSI cohorts. Regarding the lipid-lowering therapy, the NSI cohort was mostly on Rosuvastatin and, to a lesser degree, on Atorvastatin, among others, such as 3-hydroxy-3-methylglutaryl coenzyme A (HMG-CoA) reductase inhibitors or statins. The SI group was treated using alternative lipid-lowering drugs, but nine patients refused to use any other therapeutic option.

### 2.2. Plasma miRNA Profile in Patients with SI

To assess whether the plasma miRNAs were differentially expressed between the groups, we first conducted screen profiling of 179 circulating miRNAs commonly found in human plasma in ten SI and ten NSI age-matched patients (Figure 1). The established criteria for the selection of miRNA candidates were high expression levels (median Cq < 35 and detected in at least 80% of all samples) and statistical significance (*p* < 0.05). A total of 10 miRNAs, let-7c-5p, let-7d-5p, let-7f-5p, miR-128-3p, miR-186-5p, miR-30e-3p, miR-376a-3p, miR-376c-3p, miR-543 and miR-574-3p, were significantly expressed according to the selection criteria and selected for further analysis.

### 2.3. miRNAs Validation Study and Their Correlation with Clinical Parameters

We validated these ten miRNAs in 45 SI and 39 NSI individuals to confirm the diagnostic robustness to discriminate between NSI and SI patients. Our results showed that let-7c-5p, let-7d-5p, let-7f-5p, miR-376a-3p and miR-376c-3p were significantly upregulated in plasma from SI cohort versus NSI (Figure 2).

We investigated the association between the differentially expressed miRNAs and some significant clinical parameters in the SI group. No correlation was found but DLP and non-HDLc levels (Table 2).

### 2.4. Circulating miRNA as a Biological Marker of SI

We assess the ability of the differentially expressed circulating miRNAs to distinguish between SI and NSI patients using the AUC-ROC. Figure 3A shows that miR-376c-3p achieved the highest AUC with a value of 0.736 (95% CI: 0.627–0.845; *p* < 0.001), indicating a moderate performance. Let-7c-5p, let-7d-5p, let-7f-5p and miR-376a-3p display AUC values of 0.652, 0.627, 0.688 and 0.682, respectively. Afterwards, we consider the diagnostic potential of the 5-miRNA set to differentiate between SI and NSI individuals. The AUC for the combination value of our miRNA panel (let-7c-5p, let-7d-5p, let-7f-5p, miR-376a-3p and miR-376c-3p) was 0.936 (95% CI: 0.887–0.985; *p* < 0.001) (Figure 3B), improving the diagnostic ability. The sensitivity, specificity and accuracy of each miRNA and the 5-miRNA panel are shown in Table 3.

### 2.5. Combination of miRNAs, Years of Dislipidemia and Non-HDLc to Categorize SI Patients

We investigated the potential value of the miRNA candidates to discriminate between the SI vs. the NSI cohort and their association with some clinical parameters. Only years of DLP and non-HDLc levels were significantly higher in SI patients (Figure 4A,B). We assess the potential of these two clinical parameters to distinguish between SI and NSI groups. The ROC curves of these factors, DLP and non-HDLc, showed AUC values of 0.700 and 0.807, respectively, indicating a moderate performance to discriminate SI from NSI patients, whereas the combination of these two parameters achieved an AUC value of 0.844 (Table 4). We developed a multivariate model to increase the diagnostic power to discriminate between SI vs. NSI combining differentially expressed miRNAs, DLP and/or non-HDLc plasmatic levels. The diagnostic performance of the 5-miRNA panel was only slightly improved with DLP as, although the AUC value was similar, this model achieved an accuracy of 84.81% against the 82.72% shown by the 5-miRNA panel (Figure 4C,D). In contrast, the diagnostic ability of the 5-miRNA set and the non-HDLc concentration was lower than that of the 5-miRNA panel by itself, showing an AUC value of 0.855. Interestingly, the combination of the 3-miRNA panel composed of let-7f, miR-376a-3p and miR-376c-3p, and DLP plus non-HDLc reached the highest diagnostic performance with an AUC value of 0.954 and an accuracy of 89.47% (Figure 4D and Table 4). We next use the Weka data mining tool to evaluate the performance of our model using 10-fold cross-validation. We ran all classifiers in Weka and, the most successful algorithm was Ada Boost M1 achieving a mean training accuracy, sensitivity, specificity, MCC and AUC of 86%, 83.67%, 88.57%, 0.714 and 0.891, respectively, keeping higher diagnostic value than the 5-miRNA panel model (Figure 5).

### 2.6. Kyoto Encyclopedia of Genes and Genomes Pathway and Gene Ontology Enrichment Analysis

We next investigated the biological significance of the five differentially expressed miRNAs using GO enrichment and KEGG pathway analysis. We used the miRNet online software (https://www.mirnet.ca, accessed on 9 November 2021) to predict the putative targets of let-7c-5p, let-7d-5p, let-7f-5p, miR-376a-3p and miR-376c-3p. The let-7c-5p, let-7d-5p, let-7f-5p, miR-376a-3p and miR-376c-3p were associated with 516, 394, 397,112 and 84 mRNAs in the miRNet database, respectively. Interestingly, one mRNA target, IGF1R, was shared by these five miRNAs (Figure 6A,B). GO analysis using the WebGesalt (WEB-based Gene Set Analysis Toolkit) computational tool showed significant enrichment of biological processes related to response to nerve growth factor and dendrite development, cell cycle, gene silencing and response to transforming growth factor-beta (TGF-ß), among others (Figure 6, left panel). KEGG pathway analysis showed several pathways, the most significant of which are the p53 signaling pathway, Forkhead members of the O class (FoxO) signaling pathway [19,20], EGFR tyrosine kinase inhibitor resistance, and glioma (Figure 6C, right panel). Finally, disease enrichment analysis identified that these five miRNAs might be involved in leukemia, mitochondrial myopathy, lactic acidosis and diabetes mellitus (noninsulin-dependent), among others (Figure 6C, bottom panel).

## 3. Discussion

We investigated the circulating miRNA signature of a high and very high cardiovascular risk population treated with statins. We report, for the first time, a fingerprint of miRNAs (let-7c-5p, let-7d-5p, let-7f-5p, miR-376a-3p and miR-376c-3p) to discriminate between SI and NSI patients. Moreover, when we add some clinical information, this circulating miRNA set increases its accuracy remarkably.

Statins are by far the most important therapy used for the treatment and prevention of cardiovascular diseases. Statins have widely proven their profits in primary and secondary cardiovascular prevention [1,2,3]. Although usually well tolerated, statin dropout rates and non-adherence are high after the first year of therapy, 43% and 24%, respectively [21]. The main reason is SAMS which leads to neglect of the treatment and dropout [22] and dramatically increases cardiovascular morbidity and mortality [5,23].

The identification of SI patients is crucial to avoiding cardiovascular risk and reaching the aims established by the guidelines [2]. These patients need to be focused differently in order to prevent the adverse impact of statins and atherosclerosis over the years. However, the current clinical diagnosis of SI has limitations. Clinically, there are no clues to suspect who will present this condition. Myotoxicity shows very heterogeneous manners, and the phenotype that includes myalgia is the most common. SAMS affects symmetrical groups of muscles such as pelvic and shoulder girdles or limbs; muscular weakness, cramps, and, more rarely, increased creatinine kinase plasmatic levels may be considered an adverse effect [24]. Some studies have proposed various scores to assess myalgia [25,26,27]. Several pathological paths have been suggested to explain statins’ toxicity as the mitochondrial dysfunction through Co-enzyme Q10 deficiency, the reduction in beta-oxidation of fatty acids, the promotion of muscular apoptosis, drug–drug interaction, exercise, or certain genetic polymorphisms [22]. The lack of knowledge of the underlying pathological mechanism that leads to SAMS impedes any preventive measures. Thus, the difficulty reaching a clinical diagnosis, the lack of circulating markers or non-invasive tests to detect this side-effect, and the dropout of the preventive therapy leads to a crucial need for a tool that solves this problem. Some groups have related miRNAs with the effectiveness of different statins, and their pleiotropic effect on the endothelial cells [28,29,30,31]. To our knowledge, the use of circulating miRNAs as a diagnostic tool for SI detection remains unexplored.

In the present study, five plasmatic miRNAs, (let-7c-5p, let-7d-5p, let-7f-5p, miR-376a-3p and miR-376c-3p) were identified as being enriched in SI patients as compared with NSI. Moreover, together, these miRNAs showed a promising potential to discriminate between SI and NSI cohorts. Thus, let-7c-5p is involved in muscle attenuation through the TFG-β factor; its intracellular effector Smad3 is known to inhibit myogenesis and stimulate adipogenesis and myofiber lipid accumulation [32,33]. Let-7f-5p, regulated by the FoxO transcription factors family, is overexpressed in myasthenia gravis in muscle-specific tyrosine kinase antibody seropositive [34]. Myasthenia gravis is a neuromuscular autoimmune disease caused by antibodies which attack receptors at the neuromuscular junction. MiR-376a-3p, proposed as a biomarker of coronary artery disease, acts by nuclear receptor-interacting protein 1 (NRIP1) leading to the cellular apoptosis. Hence, miR-376a-3p might be related to the effect of statins reducing the risk of coronary artery disease, however, this hypothesis needs to be investigated [35]. Related to statin toxicity, others circulating miRNAs have been described as miR-499 and miR-145 [36]. In 2016, Min et al., using combined in vivo and in vitro modelling, demonstrated a release of miR-499-5p linked to muscle injury during exercise when muscular contraction induced by carbachol is combined with statins intake. Circulating levels of miR-1, miR-133a and miR-206 were significantly upregulated in patients in statin therapy who practiced endurance sports and showed muscular adverse effects [37]. Fu et al. demonstrated the statin-induced injury on the skeletal muscle through the overexpression of miR-1a by regulating the mitogen-activated kinase 1 (MKK1) pathway [38]. Some murine models have been used to demonstrate that moderate and gradual exercise may have a beneficial role on statin intake and its impact on skeletal muscle [39,40].

In the biological process analysis, we identified that these miRNAs were highly enriched with genes in the TGF-β pathway, suggesting that they are plausible candidates. Recently, has been reported that TGF-β through Smad3 transcriptional repression leads to the inhibition of myogenesis differentiation [41,42]. Myogenic differentiation and function may need several cellular signal and growth factors, but also an appropriate environment that may be impair by statins.

The genes predicted to be impacted by these five miRNAs are shown in Figure 6. All miRNAs described, except let-7f-5p, shared POTEG, BEND4, POTEM and CASTOR that are genes linked with muscular homeostasis and apoptotic processes. In this sense, CASTOR2, among other genes, has been described as a target of the proteolytic pathway of FoxO driving to muscular atrophy [43]. TGFβR1, common to all miRNAs except miR-376a-3p, has been reported to affect the muscular growth and differentiation [44]. TGFβ signaling plays a critical role in regulating muscle growth and atrophy, in both inherited and acquired myopathies [45]. It is important to note that miRNA target prediction analysis identified insulin-like growth factor (IGF1) as a target gene shared by these five miRNAs. Several studies have reported that statins regulate IGF1-R signaling at different levels [46,47,48]. AKT is a key-regulator of the synthesis and degradation muscle growth and regeneration through AKT/mTOR and FoxO transcription factors [49]. IGF1 is related to statin myotoxicity by inhibiting the IGF1/AKT pathway leading to an increased myofibrillar proteins degradation, impaired proteins synthesis. Thus, statins inhibit AKT phosphorylation activates caspases and PAPR inducing apoptosis and, most importantly, are closely associated with the inhibition of cholesterol synthesis at the level of hydroxymethylglutaryl-CoA (HMG)-CoA reductase. These findings have been demonstrated in both in vitro and in vivo models, enhancing the importance of these pathways in statin-induced myotoxicity [49]. FoxO are a family of transcription factors that are targeted by these miRNAs. They are implicated in the protein degradation and apoptosis. FoxO are related to muscular atrophy and caquexia with the use of certain chemotherapy treatments through the IGF1/AKT/FoxO pathways [50]. Finally, GO enrichment analysis revealed significant enrichment of mitochondrial myopathy-related diseases. In agreement with this, numerous studies have shown that mitochondria play an important role in statin-induced myopathies. Moreover, it has been suggested that statins may have major effects on mitochondrial function, and some of their adverse effects might be mediated through mitochondrial pathways [51,52,53]. Hence, our data suggest that these circulating miRNAs might be involved in the mechanisms of action of statins on mitochondrial function.

In this study we demonstrate that a panel of five miRNAs have a better performance in predicting patients with SI than individual miRNAs. To the best of our knowledge, these five circulating miRNAs have not been reported to be associated with SI. Moreover, we present a multiparametric model that increases the discriminative power of this diagnostic tool in clinical practice. We worked with several clinical parameters and a combination of miRNAs. Thus, our miRNAs set, let-7f-5p, miR-376a-3p and miR-376c-3p, with two clinical parameters such as DLP and non-HDLc concentration showed a high-yield diagnostic accuracy with an AUC of 0.954 and an accuracy of 89.47%. A 10-fold cross-validation evaluation to estimate the prediction performance resulted in high diagnostic accuracy (86%) with an AUC of 0.891 supporting the robustness of the multiparametric model.

Non-HDLc, a reliable marker for cardiovascular risk and residual risk, encompasses all lipoproteins that contain cholesterol and apo B, being closely related to atherogenic dyslipidemia. Thus, in any clinical context associated with insulin resistance, such as diabetes mellitus 2, metabolic syndrome or visceral obesity, it is suggested to use non-HDLc or apo B as a therapeutic aim rather than classic LDLc [54]. This highlights the importance of this non-HDLc for better control of dyslipidemia [2] and an update of the risk calculation indices has recently been published SCORE2 [55] and SCORE2-OP [56]. This update introduces different variations on the original SCORE index assessing not only cardiovascular mortality but the risk of developing a cardiovascular event. In this sense, non-HDLc has been enhanced as a major cardiovascular risk factor with the same impact as LDLc or global cholesterol. Thus, non-HDLc levels are equivalent in terms of risk to high blood pressure, diabetes mellitus or chronic kidney disease. On the other hand, DLP play a key role in ASCVD. The time of exposition to a reduced LDLc concentration is independent of the reduced mechanism used; the clinical benefit to more decreased LDLc levels, whether genetically or pharmacologically, is determinant; the lower and the earlier the better [57].

We expose a circulating miRNAs fingerprint as an accessible and helpful tool for SI diagnosis. In addition, we propose a multiparametric model, a panel of three miRNAs and two clinical parameters, that detects patients with this adverse event with the highest diagnostic value reported to date. The bioinformatics analysis leads to the biological and molecular processes present in this entity. This proposal permits a therapy-tailored strategy to avoid the dropout of a crucial therapy and offer an alternative to treat patients with high cardiovascular risk.

Our current study has several limitations. Firstly, our sample was prospectively recruited from the outpatient clinic. The size of the study sample, recruited from various Lipid Units and Cardiology Departments, did not allow us to obtain a robust multivariate logistic regression model. Furthermore, a larger sample size is needed to validate these data. As a consequence, these results should be extended and replicated to a larger population before the novel biomarkers can be routinely applied in clinical practice. Finally, even though databases registered the expression of the parental genes, we have no confirmation about the direct secretion of these circulating miRNAs into the extracellular space. Hence, the association of miRNAs with SI and all the interactions are putative. Therefore, larger confirmative studies on circulating miRNAs are needed to generate a better, reproducible prognostic signature for SI.

## 4. Materials and Methods

### 4.1. Study Population and Design

This is a multicenter prospective case-control study. Patients were recruited from three field centers (Puerta del Mar University Hospital, Cádiz; Virgen del Rocio University Hospital, Sevilla; and Reina Sofia University Hospital of Córdoba, Spain). Inclusion criteria were patients over 18 years old with a high and very high cardiovascular risk in treatment with statins referred to the Lipid Unit of these centers. We have focused on the most frequent symptom, SAMS [5]. A total of 84 consecutive subjects (Figure 7) were included and divided into two different cohorts: (i) SI patients (*n* = 39) and, (ii) patients non-intolerants to statins (NSI) (*n* = 45). SI group was defined based on the European Medicine Agency criteria [5]. Before including these patients in the study, we applied the 2019 ESC/EAS Lipid Management Guidelines, which establish the requirement to offer the patient alternative dosing such as every other day or twice a week with Atorvastatin, Pitavastatin or Rosuvastatin. All patients included dropped out of the alternative proposed due to intolerance to statins. All patients recruited were high or very high cardiovascular risk based on the classification of the cardiovascular risk grade of the ESC/EAS guidelines (Table 5) [2]. ASCVD can be documented as either clinical or unequivocal based on imaging. In the clinical branch, ASCVD includes previous acute coronary syndrome (myocardial infarction or unstable angina), stable angina, coronary revascularization procedures including percutaneous coronary intervention, coronary artery bypass graft or arterial revascularization procedures, ischemic stroke, and peripheral arterial disease. Conditions considered to explicitly document ASCVD on imaging include findings known to be predictive of clinical events, such as significant plaque on coronary angiography or coronary tomography scan (a multivessel coronary disease with two major epicardial arteries having >50% stenosis) or on carotid ultrasound.

Exclusion criteria were patients over 80 years old or with marked frailty, intensive physical activity, heavy alcohol consumption and patients who presented chronic inflammatory, autoimmune or neoplasia disease or other pharmacological therapies that interfere with the metabolic pathways of the different statins. Subjects with high training exercise workload, previous muscular symptoms, myopathy disease, high levels of creatine kinase levels or hypothyroidism were excluded from the study.

Detailed anthropometric, clinical and pharmacological information was obtained from each subject including cardiovascular risk factors and previous vascular events. All patients performed a blood test including thyroid, hepatic, kidney, lipid profiles and creatine kinase at the inclusion moment. Our institution’s ethics committee (Comité de Ética de la Investigation de Cádiz) approved the study protocol. The study was performed in full compliance with the Helsinki II Declaration. All participants provided written informed consent.

### 4.2. Blood Collection

Ten milliliters of peripheral blood were collected in K2-ethylenediaminetetraacetic acid tubes after 10 h overnight fasting and were immediately centrifuged (1500× *g*, 15 min, 4 °C). The blood was processed within 4 h after isolation. The upper layer containing plasma was divided into aliquots and stored at −80 °C until further analysis.

### 4.3. RNA Isolation

Total RNA was extracted from 200 µL of plasma using the miRNeasy Serum/Plasma Advanced Kit (Qiagen, Hilden, Germany) according to the manufacturer’s instructions. A total of 200 µL of plasma aliquots were used to obtain enough final volume. During the extraction, 3.5 μL of miRNeasy Serum/Plasma Spike-In Control (1.6 × 108 copies/μL of the *C. elegans* miR-39 miRNA mimic) were added to each sample as an internal control. RNA was eluted in 20 μL of RNase-free water.

### 4.4. MiRNA Real-Time Reverse Transcriptase-Polymerase Chain Reaction

To evaluate miRNA expression levels and profiles of SI and NSI groups, an analysis of 179 miRNA species known to be present in human serum was performed using miRCURY LNA RT Kit (Qiagen, Hilden, Germany) for reverse transcription, miRCURY LNA SYBR Green PCR Kit (Qiagen) for quantitative real-time (qRT-PCR) amplification and Human Serum/Plasma Focus, miRCURY LNA miRNA Focus PCR panels (Quiagen, Hilden, Germany) according to the manufacturer’s instructions. Raw cycle threshold (Cq) values were inter-plate calibrated using UniSp3. Cqs above 35 cycles were censored at the minimum level observed for each miRNA. As housekeeping miRNA for data normalization, we selected the most stably expressed pair of miRNAs (miR-148a-3p and let-7b-5p) as determined by the Normfinder algorithm [58]. Accordingly, each single miRNA was normalized as ∆Cq = mean (CqmiR-148a-3p and let-7b-5p)—Cq miRNA. MiRNA levels were log-transformed before being used in the statistical analyses.

### 4.5. Validation of miRNA Profiles

For the validation study, each selected miRNA candidate was quantified using miRCURY LNA miRNA Custom PCR Panels (Qiagen, Hilden, Germany). RNA was reverse transcribed using the miRCURY LNA RT Kit (Qiagen, Hilden, Germany). qRT-PCR was performed using miRCURY LNA SYBR Green PCR Kit (Qiagen, Hilden, Germany), as previously described [59] and amplification curves were evaluated with CFX Manager™ software (BioRad). The specificity of the amplification was corroborated by melting curve analysis.

### 4.6. miRNA-Gene Network Analysis

The miRNAs obtained were tested using the miRNet database (https://www.mirnet.ca, accessed on 9 November 2021) to predict the targeted genes. A database analysis to identify the biological function was performed using gene ontology (GO) enrichment analysis (http://geneontology.org/, accessed on 9 November 2021) [60] and the Kyoto Encyclopedia of Genes and Genomes (KEGG) http://www.genome.jp/kegg/, accessed on 9 November 2021) [61]. The Search Tool for the Retrieval of Interacting Genes (STRING) database (http://www.string-db.org/, accessed on 9 November 2021) [62] was used to analyze the protein–protein interaction networks.

### 4.7. KEGG and GO Term Enrichment Analysis

The miRNet online software (https://www.mirnet.ca, accessed on 9 November 2021) was used to predict the targeted genes and to construct the miRNA–mRNA regulatory network. The WebGestalt (WEB-based GEne SeT AnaLysis Toolkit) [63] computational tool was performed for KEGG pathway and GO terms analysis. GO term analysis was employed to determine the involvement of the 5 differentially expressed miRNAs in biological processes and diseases.

### 4.8. Statistical Analysis

Continuous variables are shown as mean ± standard deviation. Categorical variables are expressed as frequency and percentage of patients (%). Outliers were identified through the Rout method, using a Q = 1% [64]. The normal distribution of each variable was verified with the Shapiro–Wilk test. Intergroup comparisons of miRNAs levels were performed using non-parametric Mann–Whitney and Kruskal–Wallis rank tests for continuous variables. An analysis of differences between groups was performed using the analysis of variance. ROC curves that characterize the diagnostic performance of candidate miRNAs and logistic regression models were plotted to determine the area under the curve (AUC) and the specificity and sensitivity of the optimal cut-offs. ROC curves were generated by plotting sensitivity against 1-specificity. Data were presented as the AUC and 95% confidence intervals. The changes in *p*-values of their variables were evaluated by the Wald test and a likelihood ratio. Training was performed with 10-fold cross-validation using Waikato environment for knowledge analysis (Weka) data mining tool. The statistical software package R was used for all analyses (Team RC. R: A Language and Environment for Statistical Computing. https://www.r-project.org, accessed on 9 November 2021).

## 5. Conclusions

In conclusion, we identified for the first time a signature of five miRNAs differentially expressed in the SI population. Our multiparametric model also combined circulating miRNAs and clinical variables, and emerges as a potential clinical tool to identify patients with muscular toxic effects as a result of statins. In daily practice, this panel with accessible clinical parameters may improve the accuracy and may lead to a tailored-treatment strategy improving the cardiovascular risk and outcome of these patients.

## Figures and Tables

**Figure 1 ijms-23-08146-f001:**
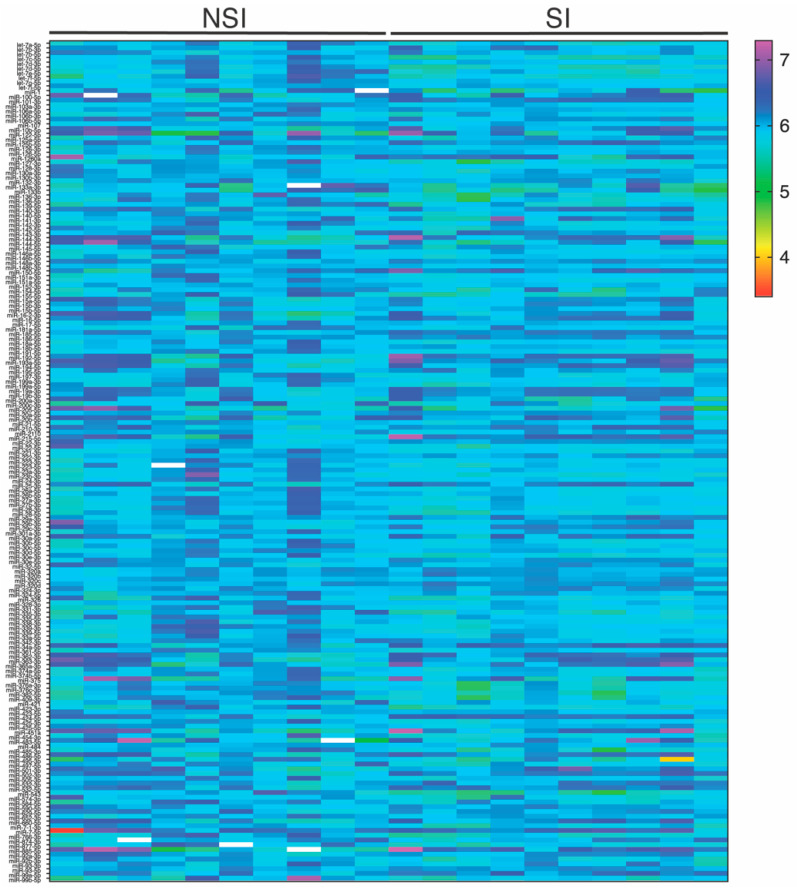
Color heatmap based on raw miRNA expression values where each column represents a patient, and each row represents a miRNA. The color scale illustrates the relative expression level of miRNAs (red and yellow represent low expression and blue and purple represent high expression). MiRNA expression levels were normalized to miR-148a-3p and let-7b-5p. MiRNA: microRNA; NSI: non-statin intolerant; SI: statin intolerant.

**Figure 2 ijms-23-08146-f002:**
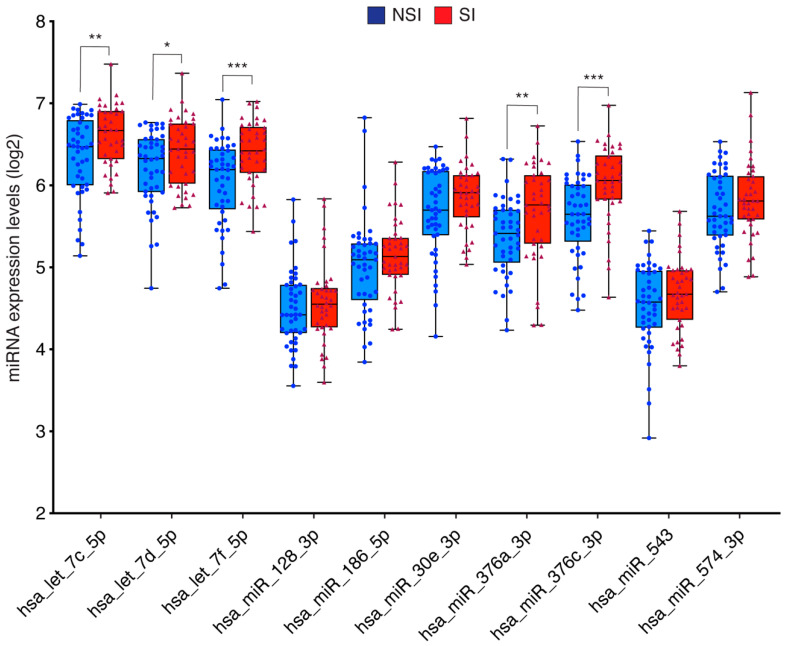
Boxplots of miRNA expression levels in NSI and SI cohorts. The analysis was carried out using qPCR. Data are presented in log2. Data represent the mean ± SEM. * *p* < 0.05, ** *p* < 0.01, *** *p* < 0.005. Error bars represent SDs. NSI, non-statin intolerant; SI: statin intolerant.

**Figure 3 ijms-23-08146-f003:**
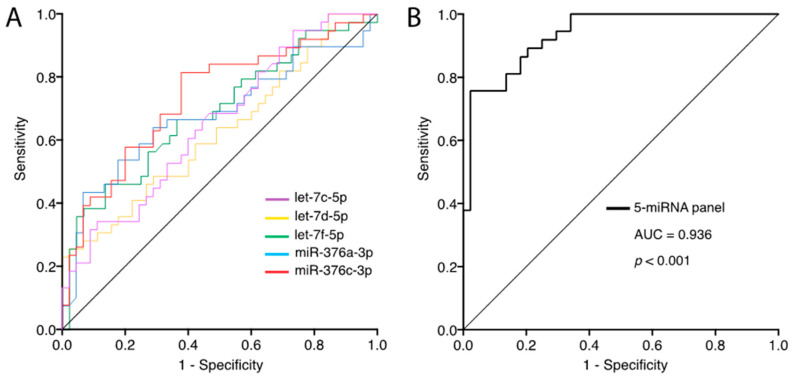
ROC curves for evaluating the predictive performance of differentially expressed miRNAs to discriminate between SI vs. NSI. (**A**) ROC curves for let-7c-5p, let-7d-5p, let-7f-5p, miR-376a-3p and miR-376c-3p. (**B**) The ROC curve of the 5-miRNA panel combination value of let-7c-5p, let-7d-5p, let-7f-5p, miR-376a-3p and miR-376c-3p. AUC: area under the curve; miRNA: microRNA; NSI: non-statin intolerant; SI: statin intolerant.

**Figure 4 ijms-23-08146-f004:**
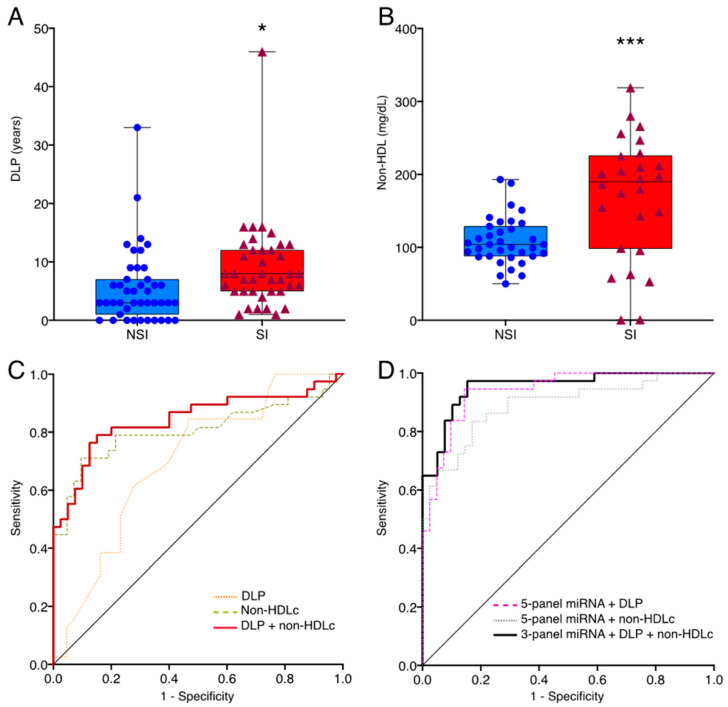
ROC curves for evaluating the predictive performance of clinical factors with differentially expressed miRNAs. (**A**) Box plot of DLP in NSI (*n* = 45) and SI (*n* = 39) subjects. (**B**) Box plot of non-HDLc levels in NSI (*n* = 45) and SI (*n* = 39) subjects. (**C**) ROC curves for each clinical parameter, DLP and non-HDLc, and for the association of DLP plus non-HDLc. (**D**) The ROC curve of the combined value of the 3-miRNA panel (let-7f-5p, miR-376a-3p and miR-376c-3p), DLP and non-HDLc plasmatic concentration. DLP: years of dyslipidemia SI: statin intolerant; NSI: non-statin intolerant; non-HDLc: non-high-density lipoprotein cholesterol. * *p* < 0.05; *** *p* < 0.005.

**Figure 5 ijms-23-08146-f005:**
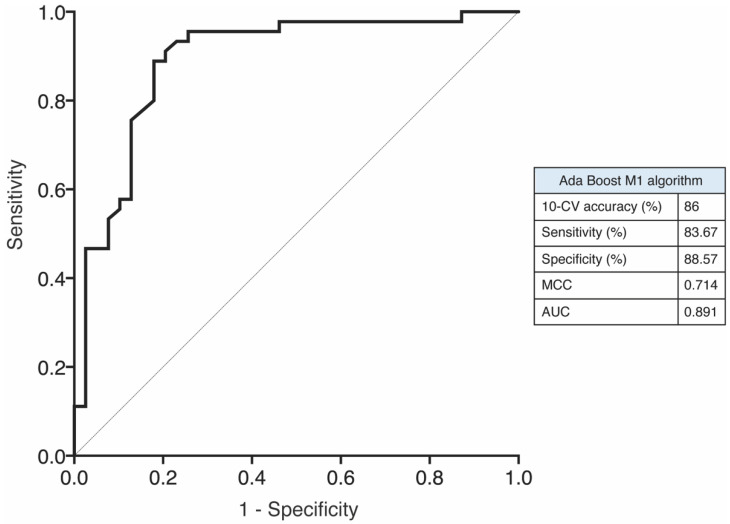
ROC curve of 10-fold cross validation test for 3-miRNA panel + DLP + non-HDLc model (Ada Boost M1).

**Figure 6 ijms-23-08146-f006:**
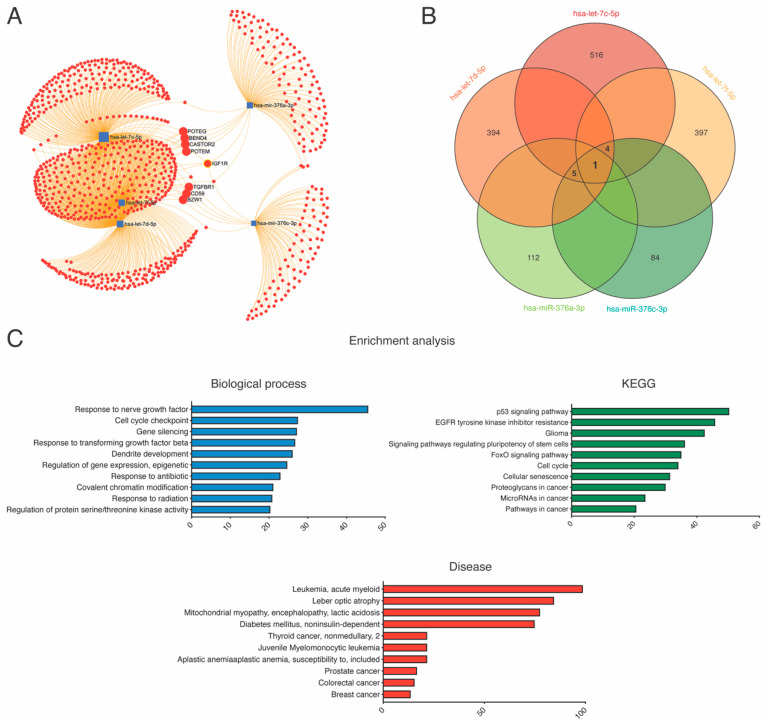
KEGG and GO analysis of differentially expressed miRNAs. (**A**) miRNA-gene network for of let-7c-5p, let-7d-5p, let-7f-5p, miR-376a-3p and miR-376c-3p. (**B**) Venn diagram showing overlap of gene targets of let-7c-5p, let-7d-5p, let-7f-5p, miR-376a-3p and miR-376c-3p. (**C**) GO and KEGG functional enrichment analysis of let-7c-5p, let-7d-5p, let-7f-5p, miR-376a-3p and miR-376c-3p.

**Figure 7 ijms-23-08146-f007:**
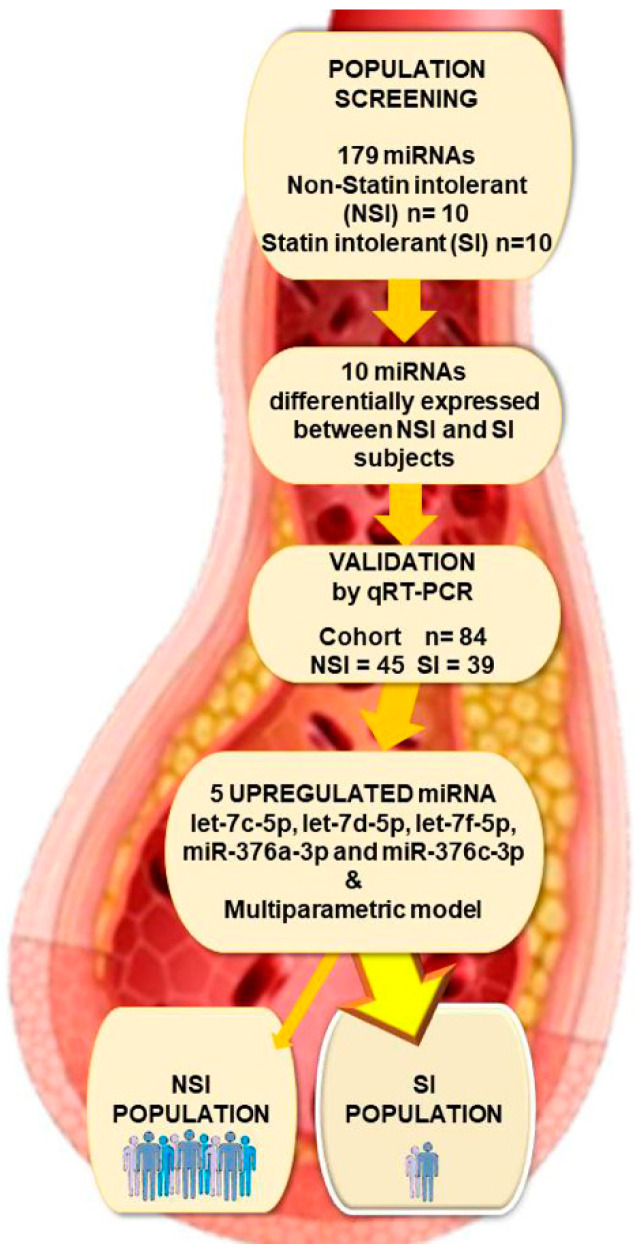
The flowchart of the study design. This figure illustrates the experimental workflow of the study including screening and validation. MiRNA: microRNA; NSI: non-statin intolerant; SI: statin intolerant.

**Table 1 ijms-23-08146-t001:** Baseline demographics, clinical characteristics, and treatment of NSI and SI population. Data are presented as mean ± SD for continuous variables and as percentage for categorical variables. The difference between NSI and SI patients was evaluated with unpaired Student *t* test ^a^, Pearson Chi-square ^b^ and Wilcoxon test ^c^. * Therapy mostly in association. NS, no significant; ACEI: angiotensin-converting enzyme inhibitors; ARB: angiotensin II receptor blockers; ASCVD: atherosclerotic cardiovascular disease; CCB: calcium channel blockers; CPK: creatine kinase; DLP: years of dyslipidemia; MDRD-4: glomerular filtration rates; non-HDLc: non-high-density lipoprotein cholesterol; NSI: non-statin intolerant; OAD: oral antidiabetic drugs; PSCK-9: Monoclonal anti-proprotein convertase subtilisin/kexin type 9; SI: statin intolerant.

Variables	NSI	SI	*p* Value
*n*	45	39	
**Demographics**			
Age (years) ^a^	66.6 ± 11.5	63.6 ± 10.9	NS
Sex (female, %) ^b^	31	61.5	0.01
DLP (years) ^c^	5.6 ± 6.3	9.1 ± 7.5	0.003
High blood pressure (%) ^b^	73	41	0.006
**Diseases**			
Diabetes Mellitus (%) ^b^	38	20.5	NS
ASCVD (%) ^b^	71	13	<0.001
Chronic kidney disease (%) ^b^	18	12.8	NS
**Analytical profile**			
Basal blood glucose ^a^	118.5 ± 36.2	110 ± 44.6	NS
Non-HDLc (mg/dL) ^a^	104.9 ± 32.7	169.9 ± 63.5	<0.001
Triglycerides (mg/dL) ^a^	151.7 ± 97.4	164 ± 93	NS
MDRD-4 (mL/min) ^a^	76.5 ± 25.6	82 ± 31.6	NS
Transaminase GOT (U/L) ^a^	23.1 ± 13	23.5 ± 8	NS
Transaminase GPT (U/L) ^a^	27.3 ± 23	24.6 ± 18.9	NS
CPK (U/L) ^a^	82.8 ± 40.4	165.9 ± 14.7	NS
**Medication**			
ACEI (%) ^b^	11	2.5	NS
ARB (%) ^b^	67	33.3	0.004
OAD (%) ^b^	33	18	NS
Insulin (%) ^b^	11	10.5	NS
Diuretic (%) ^b^	51	18	0.003
CCB (%) ^b^	31	18	NS
Beta-blockers (%) ^b^	62	15.4	<0.001
Alpha-blockers (%) ^b^	22	5	0.03
Aspirin (%) ^b^	64	18	<0.001
Atorvastatin 40 mg (%)	20	-	
Atorvastatin 80 mg (%)	15.5	-	
Rosuvastatin 10 mg (%)	17.7	-	
Rosuvastatin 20 mg (%)	35.5	-	
Pitavastatin 4mg (%)	6.6	-	
Simvastatin 40 mg (%)	4.4	-	
PCSK9 inhibitors: Evolucumab (%) ^b^	13	7.6	NS
PCSK9 inhibitors Alirocumab (%) ^b^	11	2.5	NS
Fenofibrate (%) ^b^	18	7.6	NS
Omega-3 (%)	-	5.1	
Colesevelam (%)	-	2.5	
Colestiramine (%)	-	12.8	
Armolipid plus * (%)	-	41.0	
Ezetimibe 10 mg * (%) ^b^	42	38	NS
Acenocumarol (%) ^b^	7	7.6	NS

**Table 2 ijms-23-08146-t002:** Correlation between the clinical parameter (DLP and non-HDLc) and individual microRNAs in SI subjects. DLP: years of dyslipidemia; non-HDLc: non-high-density lipoprotein cholesterol SI: statin intolerant. Coefficient significant at *p* < 0.05.

microRNAs	SI Cohort			
	DLP (Years)		Non-HDLc (mg/dL)	
	Pearson r	*p*	Pearson r	*p*
Let-7c-5p	−0.164	0.281	0.177	0.244
Let-7d-5p	−0.079	0.603	0.226	0.131
Let-7f-5p	−0.136	0.368	0.168	0.266
miR-376a-3p	−0.173	0.250	0.103	0.498
miR-376c-3p	−0.265	0.079	0.176	0.248

**Table 3 ijms-23-08146-t003:** Assessment of the potential diagnostic value of differentially expressed miRNAs and the 5-miRNA panel (let-7c-5p, let-7d-5p, let-7f-5p, miR-376a-3p and miR-376c-3p) as biomarkers to categorize statin intolerant patients. AUC: area under the curve; CI: confidence interval; miRNA: microRNA.

miRNA	AUC (95% CI)	Sensitivity %	Specificity %	Accuracy %	*p* Value
Let-7c-5p	0.652 (0.535 to 0.770)	61.70	55.56	59.04	0.017
Let-7d-5p	0.627 (0.507 to 0.747)	52.63	58.70	55.95	0.046
Let-7f-5p	0.688 (0.573 to 0.803)	60.53	64.44	62.65	0.003
miR-376a-3p	0.682 (0.563 to 0.800)	68.89	64.10	66.67	0.004
miR-376c-3p	0.736 (0.627 to 0.845)	70.45	64.10	67.47	<0.001
5-miRNA panel	0.936 (0.887 to 0.985)	81.25	84.85	82.72	<0.001

**Table 4 ijms-23-08146-t004:** Evaluation of the potential of clinical parameters (DLP and non-HDLc) and the multivariate models as SI biomarkers. AUC, area under the curve; CI, confidence interval. DLP: years of dislipidemia; miRNA: microRNA; non-HDLc: non-high-density lipoprotein cholesterol.

Multiparametric Model	AUC (95% CI)	Sensitivity %	Specificity %	Accuracy %	*p* Value
DLP (years)	0.700 (0.587 to 0.814)	57.89	60.00	58.54	0.017
Non-HDLc (mg/dL)	0.807 (0.703 to 0.911)	77.08	84.38	80.00	<0.001
DLP + non-HDLc	0.844 (0.751 to 0.937)	79.55	85.29	82.05	<0.001
5-miRNA panel + DLP	0.940 (0.892 to 0.989)	85.71	83.78	84.81	<0.001
5-miRNA panel + non-HDLc	0.889 (0.814 to 0.964)	85.00	81.08	83.12	<0.001
3-miRNA panel + DLP + non-HDLc	0.954 (0.911 to 0.998)	89.74	89.19	89.47	<0.001

**Table 5 ijms-23-08146-t005:** Cardiovascular risk criteria based on the 2019 ESC/EAS guidelines used to the recruitment of patients [2]. ASCVD: atherosclerotic cardiovascular disease; DM: diabetes mellitus; CKD: chronic kidney disease; CVD: cardiovascular disease; eGFR: estimated glomerular filtration rate; FH: familial hypercholesterolemia; SCORE: Systematic coronary risk estimation. Adapted from Ref. [2]. Copyright 2019. The European Society of Cardiology and the European Atherosclerosis Association 2019.

Very High CVD Risk	High CVD Risk
Presence of ASCVD clinically/imaging.	Total cholesterol over 310 mg/dL, LDLc over 190 mg/dL or blood pressure ≥ 180/110 mmHg.
DM patients with target organ damage or at least three major risk factor or early onset of DM type 1 with a length over 20 years.	DM patients without target organ damage. Over 10 years with DM.
Severe CKD (eGFR < 30 mL/min/1.73 m^2^).	Moderate CKD (eGFR = 30–39 mL/min/1.73 m^2^).
A calculated SCORE ≥ 10% for 10 years risk of fatal CVD.	A calculated SCORE ≥ 5% and < 10% for 10 years’ risk of fatal CVD.
FH with a ASCVD or with another major risk factor.	FH without any other major risk factor.

## Data Availability

Data transparency is guaranteed. The datasets generated during and/or analyzed during the current study are available from the corresponding author on reasonable request. We used various software for functional enrichment and statistical analysis. All of them are cited in our manuscript.

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
