# Peer review of "A microRNA Signature for the Diagnosis of Statins Intolerance"

_ijms, 2022, doi:10.3390/ijms23158146_

Round 1

Reviewer 1 Report

This paper studied associated mi-RNAs with statins intolerance. 

I think the results seem very interesting, and the paper is organized well in general. 

However, I would like to ask a question about your prediction model construction. In your model construction and validation with AUC and the other prediction measures, I could not find data division step such as training, test, and validation data set. If you omitted the explanation of the process, please add them into the paper. If your AUC and the other values are from only training data set, it might be overestimated. 

In addition, as a minor comment, please add a reference for rout method in line 175. 

Author Response

Dear Reviewer,

Please find enclosed our manuscript entitled: ‘A miRNAs signature for the diagnosis of statins intolerance’ which we are submitting for your consideration as an original research article. We thank the reviewer for all the comments and the time invested in our manuscript.

We attach all their comments point by point all the changes performed following the reviewer’s suggestions

R1.Q1:

However, I would like to ask a question about your prediction model construction. In your model construction and validation with AUC and the other prediction measures, I could not find data division step such as training, test, and validation data set. If you omitted the explanation of the process, please add them into the paper. If your AUC and the other values are from only training data set, it might be overestimated.

R1.A1: First of all, we would like to thank the reviewer for his/her comments on our manuscript which will certainly help to improve it in the revised version of the manuscript.

Along the construction of our model, we did not perform any training tests as the sample size was limited. However, following your suggestion, we have evaluated the real performance of our model with 10-fold cross-validation using the Weka tool. All the classifiers in Weka were run to further evaluate our model. Data obtained  support the robustness of our model as high accuracies were obtained in multiple of them (see table below). We have added this analysis and the model with the highest accuracy in the revised version of the manuscript. We have also included in the “Methods Section” and the “Results Section”, highlighted in yellow, additional information for a better understanding for the readers.

 “We next use the Weka data mining tool to evaluate the performance of our model using 10-fold cross-validation. We ran all classifiers in Weka and, the most successful algorithm was Ada Boost M1 achieving a mean training accuracy, sensitivity, specificity, MCC and AUC of 86%, 83.67%, 88.57%, 0.714 and 0.891, respectively, keeping higher diagnostic value than the 5-miRNA panel model (Figure 6).

We would like to highlight that this model will be validated in a bigger population in a second phase of the project

Algorithm

TP Rate

FP Rate

Accuracy

Recall

F-Measure

MCC

ROC Area

PRC Area

meta.AdaBoostM1

0.857

0.151

0.859

0.857

0.856

0.714

0.891

0.884

meta.Bagging

0.821

0.189

0.825

0.821

0.82

0.643

0.855

0.845

function.SGD

0.81

0.203

0.814

0.81

0.808

0.62

0.803

0.749

meta.MultiClassClassifierUpdateable

0.81

0.203

0.814

0.81

0.808

0.62

0.803

0.749

bayes.NaiveBayesUpdateable

0.81

0.196

0.81

0.81

0.809

0.616

0.815

0.779

trees.HoeffdingTree

0.81

0.196

0.81

0.81

0.809

0.616

0.816

0.781

trees.RandomForest

0.81

0.196

0.81

0.81

0.809

0.616

0.856

0.845

functions.SMO

0.798

0.22

0.809

0.798

0.794

0.601

0.789

0.736

rules.DecisionTable

0.798

0.22

0.809

0.798

0.794

0.601

0.807

0.785

functions.SimpleLogistic

0.798

0.213

0.8

0.798

0.796

0.594

0.852

0.848

trees.LMT

0.798

0.213

0.8

0.798

0.796

0.594

0.852

0.848

R1. Q2:

In addition, as a minor comment, please add a reference for rout method in line 175.

R1. A2: Indeed, we apologize. We have added the mentioned reference for rout method in the revised version of the manuscript.

  1. Motulsky, H. J.; Brown, R. E. Detecting outliers when fitting data with nonlinear regression - a new method based on robust nonlinear regression and the false discovery rate. BMC Bioinformatics. 2006, 7(123). doi: 10.1186/1471-2105-7-123.

Also, we have checked the English language and spelling. 

Reviewer 2 Report

Manuscript ID: ijms-1807915

Dear Authors,

 I have received your manuscript entitled “A miRNAs signature for the diagnosis of statins intolerance” to be considered for publication in International Journal of Molecular Sciences. This study identified possible new plasma microRNA (miRNA) biomarkers (let-7c-5p, let-7d-5p, let-7f-5p, miR-376a-3p and miR-376c-3p) in investigated intolerant to statins (SI) population. The differentiation was performed between group of SI and non-statin intolerant (NSI) population. It was screened and statistically evaluated 179-differentially expressed miRNAs used novel advanced technologies for that. It was selected the five miRNA which probably may personalize dyslipidemia treatment in patients with cardiovascular risk.

 My comments:

 What group of statins did they use in the NSI ? It is known that atorvastatin repressed selected miRNAs in hypercholesterolemic subjects.

What statin replacement drugs were they using in the SI group? 

You are showing an analytical profile for 84 patients (39 SI patients and 45 NSI patients as the control group), but how many patients were met "criteria for the selection of miRNA candidates ... high expression levels". 

Have the used drugs been checked to affect miRNAs level ?

As you wrote “these 5 miRNAs might be involved in leukemia, mitochondrial myopathy, lactic acidosis and diabetes mellitus (noninsulin-dependent), among others”, so, are not specific for differentiation between group of SI and NSI.

Few suggestions you can find in marked manuscript pdf version .

I hope you will find the comments and marks helpful for the improvement of the manuscript.

Reviewer

Author Response

Dear Reviewers ,

Please find enclosed our manuscript entitled: ‘A miRNAs signature for the diagnosis of statins intolerance’ which we are submitting for your consideration as an original research article. We thank the reviewer for all the comments and the time invested in our manuscript.

We attach all their comments point by point all the changes performed following the reviewer’s suggestions

Reviewer 3 Report

1. General comments:

- Subject of interest

- Original study

- Well structured and written manuscript

2. Specific comments:

- What definition of statin intolerance for the inclusion of the studied patients?

- Presentation of the symptoms, clinical signas and biologica data for statin intolerence for the two groups of patients

- History of statin used in the two groups (type of molecule, duration, posology...)? History of alcohol intake?

Author Response

Dear Reviewers ,

Please find enclosed our manuscript entitled: ‘A miRNAs signature for the diagnosis of statins intolerance’ which we are submitting for your consideration as an original research article. We thank the reviewer for all the comments and the time invested in our manuscript.

We attach all their comments point by point all the changes performed following the reviewer’s suggestions. 

Round 2

Reviewer 2 Report

Thank you for the comprehensive information.